# Methicillin-resistant and methicillin-sensitive *Staphylococcus aureus* isolates from skin and nares of Brazilian children with atopic dermatitis demonstrate high level of clonal diversity

**Lorrayne Cardoso Guimarães**[1], **Maria Isabella de Menezes Macedo Assunção**[1], **Tamara Lopes Rocha de Oliveira**[1], **Fernanda Sampaio Cavalcante**[2], **Simone Saintive**[3], **Eliane de Dios Abad**[3], **Ekaterini Simoes Goudouris**[4], **Evandro Alves do Prado**[4], **Dennis de Carvalho Ferreira**[5,6,7], **Kátia Regina Netto dos Santos**[1]*

1 Departamento de Microbiologia Médica, Instituto de Microbiologia Paulo de Góes, Universidade Federal do Rio de Janeiro, Rio de Janeiro, Rio de Janeiro, Brazil, 2 Departamento de Clínica Médica, Instituto de Ciências Médicas, Universidade Federal do Rio de Janeiro, Campus Macaé, Macaé, Rio de Janeiro, Brazil, 3 Ambulatório de Dermatologia Pediátrica, Instituto de Puericultura e Pediatria Martagão Gesteira, Universidade Federal do Rio de Janeiro, Rio de Janeiro, Rio de Janeiro, Brazil, 4 Ambulatório de Alergia Pediátrica, Instituto de Puericultura e Pediatria Martagão Gesteira, Universidade Federal do Rio de Janeiro, Rio de Janeiro, Rio de Janeiro, Brazil, 5 Faculdade de Odontologia, Universidade Estácio de Sá, Rio de Janeiro, Rio de Janeiro, Brazil, 6 Faculdade de Odontologia, Universidade Veiga de Almeida, Rio de Janeiro, Rio de Janeiro, Brazil, 7 Faculdade de Enfermagem, Departamento de Fundamentos de Enfermagem, Universidade do Estado do Rio de Janeiro, Rio de Janeiro, Rio de Janeiro, Brazil

* santoskrn@micro.ufrj.br

## Abstract

### Background

Atopic dermatitis (AD) primarily affects the pediatric population, which is highly colonized by *S. aureus*. However, little is known about the genetic features of this microorganism and other staphylococcal species that colonize AD patients.

### Objective

This study aimed to characterize *Staphylococcus* spp. isolated from the nares and skin (with and without lesion) of 30 AD and 12 non-AD Brazilian children.

### Methods

Skin and nasal swabs were cultured onto mannitol salt agar, and bacterial colonies were counted and identified by matrix assisted laser desorption ionization time of flight mass spectrometry and polymerase chain reaction (PCR). Antimicrobial susceptibility was evaluated by phenotypic and genotypic tests. In *S. aureus* isolates, Panton-Valentine leukocidin genes were detected by PCR, and their clonality was assessed by pulsed-field gel electrophoresis and multilocus sequence typing.

**Data Availability Statement:** All relevant data are within the manuscript and its Supporting information files.

**Funding:** Author who received each award: a) KRNS This study was supported by Brazilian grants from Fundação Carlos Chagas Filho de Amparo à Pesquisa do Estado do Rio de Janeiro (FAPERJ, grants #E-26/202.592/2019, #E-26/010.000172/2016, #E-26/010.001463/2019, #E-26/010.101056/2018; #E-26/211.554/2019 (Projeto REDES); #E-26/201.071/2020; and #E-26/211.284/2021), Conselho Nacional de Desenvolvimento Científico e Tecnológico (CNPq, grants #307594/2021-1 and #26.210.875/2016), and Coordenação de Aperfeiçoamento Pessoal de Nível Superior – Brasil (CAPES)– Finance Code 001. B) DCF This study was supported by Brazilian grants from Fundação Carlos Chagas Filho de Amparo à Pesquisa do Estado do Rio de Janeiro (FAPERJ, grants #E-26/203.296/2017; and #E-26/201.454/2021). The funders had no role in study design, data collection and analysis, decision to publish, or preparation of the manuscript.

**Competing interests:** The authors have declared that no competing interests exist.

## Results

*S. aureus* was more prevalent in the nares ($P = 0.005$) and lesional skin ($P = 0.0002$) of children with AD, while *S. hominis* was more frequent in the skin of non-AD children ($P < 0.0001$). All children in the study, except one from each group, were colonized by methicillin-resistant coagulase-negative *Staphylococcus* and 24% by methicillin-resistant *S. aureus*. Despite the great clonal diversity of *S. aureus* (18 sequence types identified), most AD children (74.1%) were colonized by the same genotype in both niches.

## Conclusion

High colonization by polyclonal *S. aureus* isolates was found among children with AD, while *S. hominis* was more frequent among non-AD children. The high prevalence of methicillin-resistant staphylococcal isolates highlights the importance of continued surveillance, especially when considering empiric antibiotic therapy for the treatment of skin infections in these patients.

## Introduction

Atopic dermatitis (AD) is an inflammatory condition that primarily affects the pediatric population. Clinical signs and symptoms include eczema, sleep disorders, food allergies, and others. Genetic, immunological, environmental, and microbiological factors are associated with AD pathogenesis [1]. In this regard, a high abundance of *Staphylococcus spp.*, especially *S. aureus*, is observed in the skin of patients with AD compared to individuals without the disease [2].

The contribution of *S. aureus* to AD pathogenesis is complex, as dysfunction of the epithelial barrier and immune system favors its growth, and the presence of this pathogen in the skin, in turn, favors the dysfunction of the epithelial barrier and the immune system [3]. The presence of virulence and toxin genes has been reported among staphylococcal isolates from AD, which may be associated with triggering, sustaining, and amplifying the inflammatory process [4]. Panton-Valentine leukocidin (PVL), for instance, is associated with necrotic skin infections and necrotizing pneumonia, and it was detected in 75% of *S. aureus* isolates from nares and infected skin of AD children in a study conducted by our group [5]. In addition, colonization by community methicillin-resistant lineages associated with high virulent potential has been reported, such as from the clonal complexes (CC) 1 and 30, carrying SCC*mec* IV, PVL, and other virulence genes [5, 6].

Unlike studies regarding *S. aureus*, those addressing the role of coagulase-negative *Staphylococcus* (CoNS) in AD are still scarce. In a prospective birth cohort study, it was observed that the prevalence of *S. hominis* in infants who developed AD was lower than in those that did not develop the disease [7]. Nakatsuji et al. (2017) reported that a group of American adults with AD had a lower colonization rate by CoNS isolates capable of inhibiting *S. aureus* growth than adults without AD [8]. It has been also reported a high expression of a protease produced by *S. epidermidis*, associated with skin damage in AD patients, which could exacerbate the disease [9]. Thus, the exact role of CoNS in the altered AD skin is not yet known.

Although *S. aureus* highly colonizes the skin and nares of AD children, there is a lack of studies evaluating colonization by other *Staphylococcus* species. Furthermore, little is known about methicillin-resistance and clonal lineages of *S. aureus* isolated from these individuals, even though antimicrobial resistance is of concern as these patients tend to have recurrent infections, and some clonal lineages are associated with more virulent traits [4, 5]. Therefore, this research

aimed to evaluate colonization by staphylococcal species on the nares and skin of AD and non-AD Brazilian children and to characterize their methicillin resistance. Aspects associated with virulence and clonality were also evaluated for *S. aureus* isolates. The relevance of this study is due to the fact that it is, to the best of our knowledge, the first to describe differences in the cutaneous and nasal staphylococcal community, colonization by diverse clonal lineages of *S. aureus*, and high prevalence of methicillin-resistant staphylococci in Brazilian AD children.

## Materials and methods

### Ethics approval and study design

This was a cross-sectional, comparative, and clinical-laboratorial study conducted in a pediatric dermatology outpatient clinic of a tertiary public hospital in Rio de Janeiro, Brazil, between June 2018 and November 2019. It was approved by the ethics committee of the Instituto de Puericultura e Pediatria Martagão Gesteira (IPPMG) (N˚3493662).

The research included 42 children aged 2 to 10 years: 30 diagnosed with AD according to the Hanifin & Rajka (1980) criteria [10] and 12 siblings of AD children without the disease. Disease severity was determined using the Scoring Atopic Dermatitis (SCORAD) index [11] assessed on the day of sample collection. Only AD children with an AD lesion in at least one side of the antecubital fossa were included in the study. Exclusion criteria for both groups were: presentation of any other dermatological or infectious condition; being hospitalized in the past 6 months; use of systemic immunosuppressors in the past 2 months; and use of topical or oral antimicrobials 7 days prior to sample collection. Both groups were instructed to avoid washing their arms as well as using moisturizers or topical products 8 h prior to sample collection. All the legal guardians and/or patients provided a written informed consent and/or assent form to participate in this study.

### Sample collection and bacterial identification

Sterile swabs pre-moistened in saline solution were carefully rubbed against the antecubital fossa skin of the AD and non-AD children in a delimited area of 9 $cm^2$ for 30 s. This procedure was repeated 2 cm above the AD lesion (healthy skin region). We were unable to collect a swab from the non-lesional skin of one AD child. A pre-moistened swab was also rubbed against each of the anterior nares in both groups. All the swabs were suspended in 1 ml 1X phosphate-buffered saline (Laborclin; Pinhais, PN, Brazil) with 0.1% TritonX-100 (Sigma Chemical Company; Saint Louis, MO, USA) and processed within 4 h of collection. The swabs were vigorously vortexed, and the suspension contents were inoculated on mannitol salt agar (Becton, Dickinson and Company; Sparks, MD, USA) and incubated at 35 ˚C for 48 h. Mannitol fermenting (MSA+) and non-fermenting (MSA-) colonies were counted and those with distinct traits were selected from the plates of each clinical specimen. Bacterial identification was performed by matrix assisted laser desorption ionization time of flight mass spectrometry (MAL-DI-TOF-MS) (Bruker Daltonics; Germany) using MALDI Biotyper software version 7.0 (Bruker Daltonics) and polymerase chain reaction (PCR) (S1 Table) [12–16] for those with a score <2.0 (probable genus identification). All isolates were stored at -20 ˚C in trypticase soy broth (Becton, Dickinson and Company; Sparks, MD, USA) with 20% glycerol (Proquimios Comercio e Indústria; Rio de Janeiro, RJ, Brazil).

### Screening for antimicrobial resistance

Methicillin susceptibility was determined using cefoxitin disk-diffusion test (Oxoid; Cambrigde, UK) for all *Staphylococcus* isolates according to Clinical Laboratory Standard Institute

(CLSI) recommendations [17]. For *S. aureus* isolates, the susceptibility to mupirocin, clinda-mycin, and sulfamethoxazole-trimethoprim (SXT) (Oxoid; Cambrigde, UK) was also evaluated. Minimal inhibitory concentrations (MIC) for oxacillin and vancomycin (Sigma Chemical Company, Saint Louis, MO, USA) were carried out by the broth microdilution method for methicillin-resistant *S. aureus* (MRSA) isolates [17].

### Detection of PVL and *mecA* genes, SCC*mec* typing

DNA was extracted according to Pitcher, Sauders & Owen (1989) [18]. PCR was conducted to detect the *mecA* gene in all *Staphylococcus* isolates [19] and the PVL genes in *S. aureus* isolates [20]. MRSA isolates had their SCC*mec* typed by multiplex-PCR [21].

### Clonality assessment by PFGE and MLST

Pulsed-field gel electrophoresis (PFGE) was performed to determine the clonal lineage of *S. aureus* isolated from lesional skin in the AD group, healthy skin in the non-AD group, and nares in both groups using the restriction enzyme *Sma*I (New England Biolabs; Ipswich, England) [22]. The restriction enzymes *Sma*I and *Apa*I (New England Biolabs; Ipswich, England) were used for isolates that were not cut by *Sma*I alone. PFGE fingerprints were analyzed with BioNumerics 7.6.3 software (Applied Maths, Biomérieux; Sint-Martens-Latem, Belgium). Isolates that had a similarity coefficient > 80% were assigned as the same genotype [23]. Multilocus sequence typing (MLST) was also performed for representative isolates of each genotype [24, 25].

### Statistical analysis

Data were analyzed using GraphPad Prism software version 8.01 (Prism, GraphPad Software, San Diego, CA, USA). The Kolmogorov-Smirnov and Shapiro-Wilk tests were used to assess whether the colony count had a normal distribution. The Kruskal-Wallis's test followed by Dunn's test for multiple comparisons were used to evaluate the colony count. The Chi-square and Fisher's exact tests were performed for the other analysis. Results were considered statistically significant when $P < 0.05$.

## Results

### Characteristics of the groups

The children in the AD group had a mean age of six years, the majority was female (19; 63%). On the other hand, the non-AD group had a mean age of 4.75 years, with an equal distribution of male and female individuals. Most AD children presented moderate SCORAD (20; 67%), followed by mild (6; 20%) and severe (4; 13%) cases.

### Nasal and skin colonization by *Staphylococcus* spp

MSA+, possibly *S. aureus*, and MSA- colonies were counted for skin specimens. MSA+ count from the lesional AD skin was significantly higher than the MSA+ count from skin without lesion (adjusted *P* value = 0.035), the MSA- count from the skin without lesion (adjusted *P* value = 0.0006) and both MSA+ and MSA- from the skin of non-AD children (adjusted *P* values = 0.0001 and 0.023, respectively) (Fig 1).

All 30 (100%) AD children presented *S. aureus* on the lesional skin, while 24 (82.7%) had this species on the skin without lesion (*P* = 0.024) (Fig 2a). Only one AD child was not a nasal carrier of *S. aureus*, but this microorganism was detected in both skin sites (S2 Table). In contrast, in the group without AD, there was a lower prevalence of *S. aureus* colonization, both in

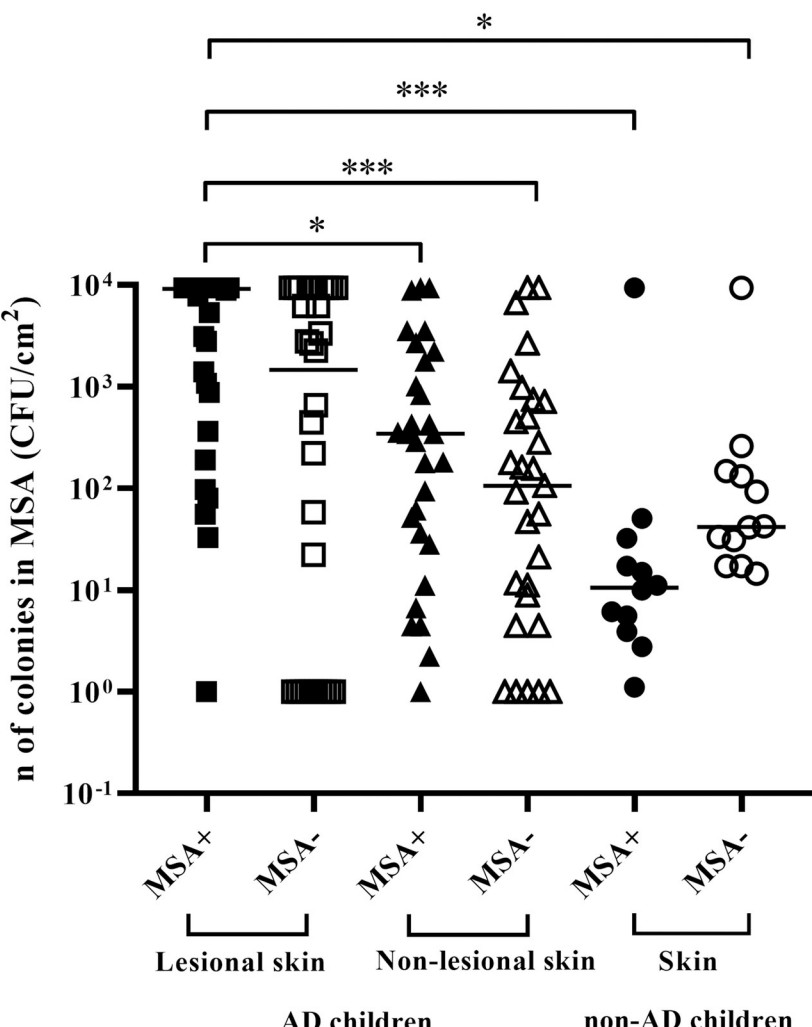

**Fig 1. Mannitol fermenting and non-fermenting colony count from skin sites of children with and without atopic dermatitis.** AD- Atopic dermatitis; $CFU/cm^2$- Colony forming units per square centimeter; MSA- Mannitol salt agar; MSA+- Mannitol fermenting colonies; MSA- Mannitol non-fermenting colonies; n- Number. The horizontal bars indicate the median count for each group. The symbols * and *** indicate, respectively: $P < 0.05$ and $P < 0.001$.

the skin, 50% ($P = 0.052$ when compared to the AD non-lesional skin; $P = 0.0002$ when compared to the AD lesional skin) and in the nostrils, 58.3% ($P = 0.005$) (Fig 2a and 2b, Table 1).

In both groups, *S. aureus* and *S. epidermidis* were the most frequently isolated species from nares (Table 1). These species were also the most frequently found in skin from AD children, while *S. epidermidis* and *S. hominis* were the prevalent species detected in this niche in the non-AD group. Colonization by *S. hominis* in the group without AD was significantly higher than in the AD group in both skin with ($P < 0.0001$) and without lesion ($P = 0.006$) and nares ($P = 0.004$) (Fig 2a and 2b). Nasal colonization by *S. saprophyticus* was also more frequent in children without AD ($P = 0.004$) (Fig 2b).

*Staphylococcus aureus* alone was more frequently detected in lesional skin than in AD non-lesional skin ($P = 0.030$) (Tables 1 and 2 and S2 Table). In fact, in three (10%) AD children, with moderate SCORAD, only *S. aureus* was recovered from both skin sites, whereas in eight (26.7%) this occurred only for lesional skin (S2 Table). In addition, three of the four children

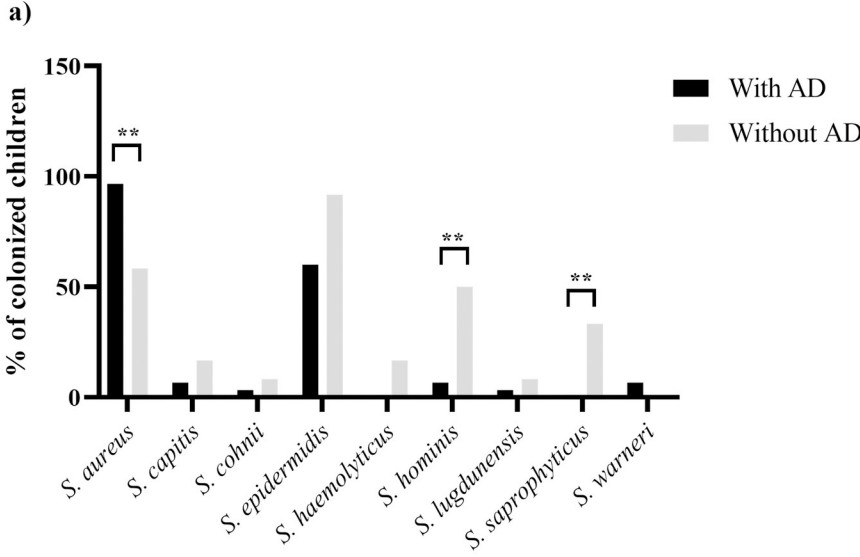

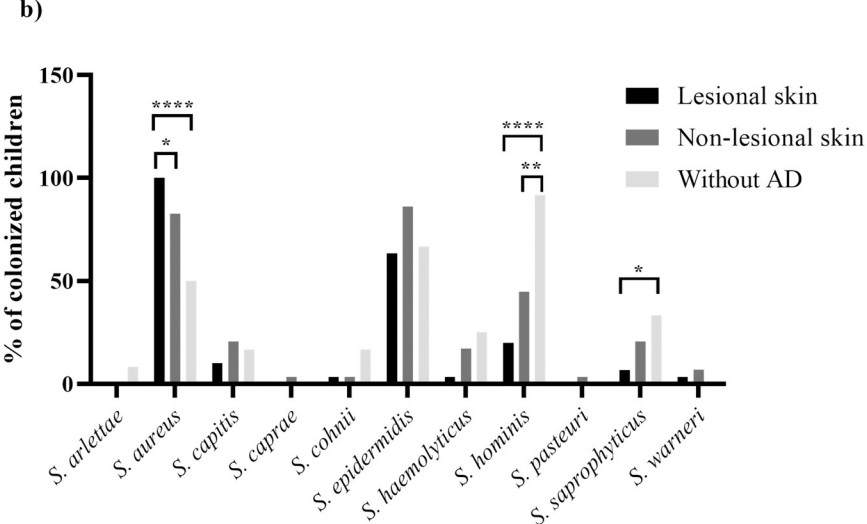

**Fig 2. Skin and nasal colonization by *Staphylococcus* spp. in 30 children with atopic dermatitis and 12 without the disease.** a) Skin colonization; b) Nasal colonization. AD- Atopic dermatitis. Collection of a non-lesional skin swab was not performed on a child in the atopic dermatitis group, and therefore the calculation of the percentage of this isolation site was performed for 29 children. The symbols *, **, and **** indicate, respectively: $P < 0.05$, $P < 0.01$, and $P < 0.0001$.

with severe AD only had *S. aureus* detected in the lesional skin. In general, most children with and without AD were colonized by three staphylococcal species in each investigated site (Table 2 and S2 Table). Furthermore, colonization by CoNS on the skin without lesion was higher than on the lesional skin among AD children ($P = 0.030$).

**Table 1. Distribution of the staphylococcal isolates according to the species, isolation sites, and methicillin susceptibility.**

| Staphylococcalspecies | N (%) of isolates | | | | | | | | |
|---|---|---|---|---|---|---|---|---|---|
| | AD children | | | | | Non-AD children | | | |
| | (n = 203) | | | | | (n = 73) | | | |
| | Nares | Lesional skin | Non-lesional skin | Total | Methicillin-resistant | Nares | Skin | Total | Methicillin-resistant |
| | (n = 55) | (n = 63) | (n = 85) | (n = 203) | (n = 90) | (n = 34) | (n = 39) | (n = 73) | (n = 37) |
| *S. aureus* | 29 (52.7) | 30 (47.6) | 24 (28.2) | 83 (40.1) | 20 (24.1) | 7 (20.6) | 6 (15.4) | 13 (17.8) | 2 (15.4) |
| *S. capitis* | 2 (3.6) | 3 (4.8) | 6 (7.1) | 11 (5.4) | 0 (0) | 2 (5.9) | 3 (7.7) | 5 (6.8) | 0 (0) |
| *S. epidermidis* | 18 (32.7) | 19 (30.2) | 26 (30.6) | 63 (31.0) | 45 (71.4) | 11 (32.3) | 9 (23.1) | 20 (27.4) | 16 (80) |
| *S. haemolyticus* | 0 (0) | 1 (1.6) | 5 (5.9) | 6 (2.9) | 4 (66.7) | 2 (5.9) | 4 (10.2) | 6 (8.2) | 6 (100) |
| *S. hominis* | 2 (3.6) | 6 (9.5) | 13 (15.3) | 21 (10.3) | 14 (66.7) | 6 (17.6) | 11 (28.2) | 17 (23.3) | 9 (52.9) |
| *S. saprophyticus* | 0 (0) | 2 (3.2) | 6 (7.1) | 8 (3.9) | 4 (50) | 4 (11.8) | 3 (7.7) | 7 (9.6) | 0 (0) |
| Another CoNS[a] | 4 (7.3) | 2 (3.2) | 5 (5.9) | 11 (5.4) | 3 (27.3) | 2 (5.9) | 3 (7.7) | 5 (6.8) | 4 (80) |

AD- Atopic dermatitis; CoNS- Coagulase-negative *Staphylococcus*; n- Number.

[a]Another CoNS- *S. arlettae*, *S. caprae*, *S. cohnii*, *S. lugdunensis*, *S. pasteuri* or *S. warneri*.

## Antimicrobial susceptibility and PVL genes

Ninety-six *S. aureus* isolates were evaluated (83 from the AD group and 13 from the non-AD group) (Table 1). Methicillin resistance was detected in 22.9% of *S. aureus* isolates colonizing 30% of AD children and 8% of non-AD children ($P = 0.233$). All MRSA isolates presented SCC*mec* IV, and oxacillin and vancomycin MICs ranged from 16 to 128 mg/L and 0.5 to 2 mg/L, respectively. All isolates were sensitive to mupirocin and clindamycin. Five (5.2%) isolates with reduced susceptibility to SXT (one intermediate resistant and four resistant) were detected in the AD group, colonizing three children.

One hundred eighty CoNS isolates were evaluated for methicillin susceptibility (120 from the AD group and 60 from the non-AD children), mainly *S. epidermidis* (46.1%) and *S. hominis* (21.1%) (Table 1). About 60% of the CoNS isolates were methicillin-resistant in both AD and non-AD groups, with more than 20% of isolates per species being resistant to methicillin

**Table 2. Major staphylococcal species associations identified colonizing the skin of children with and without atopic dermatitis.**

| Staphylococcal species associations | N (%) of children | | | | | |
|---|---|---|---|---|---|---|
| | AD (n = 30) | | | P-value[b] | Non-AD (n = 12) | |
| | Nares | Lesional skin | Non-lesional skin[a] | | Nares | Skin |
| Only *S. aureus* | 7 (23.3) | 11 (36.7) | 3 (10.3) | **0.03** | 0 (0) | 0 (0) |
| Only *S. epidermidis* | 0 | 0 (0) | 0 (0) | na | 0 | 0 (0) |
| Only *S. hominis* | 0 | 0 (0) | 0 (0) | na | 0 | 1 (8.3) |
| *S. aureus* + *S. epidermidis* | 9 (30) | 6 (20) | 5 (17.2) | >0.999 | 2 (16.6) | 0 (0) |
| *S. aureus* + *S. hominis* | 0 (0) | 0 (0) | 0 (0) | na | 0 (0) | 2 (16.7) |
| *S. aureus* + *S. epidermidis* + *S. hominis* | 2 (6.7) | 5 (16.7) | 5 (17.2) | >0.999 | 1 (8.3) | 3 (25) |
| *S. aureus* + *S. epidermidis* + another CoNS[c] | 2 (6.7) | 7 (23.3) | 2 (6.9) | 0.145 | 1 (8.3) | 1 (8.3) |

AD- Atopic dermatitis; CoNS- Coagulase-negative *Staphylococcus*; n- Number; na- Not applicable.

[a]Collection of a non-lesional skin swab was not performed on a child in the atopic dermatitis group, and therefore the calculation of the percentage of this isolation site was performed for 29 children.

[b]P-value in relation to colonization in skin with and without lesion from children with atopic dermatitis (calculation used 30 children with lesional skin and 29 with non-lesional skin samples).

[c]CoNS species other than *S. hominis*.

(excepting *S. capitis*, which was sensitive). All children in the study, except one from each group, were colonized by methicillin-resistant coagulase-negative *Staphylococcus* (MR-CoNS).

Nineteen (22.9%) *S. aureus* isolates were PVL+, colonizing nine (30%) AD children and three (25%) non-AD children.

## Clonal profiles of *S. aureus* isolates from skin and nares

There was a high diversity of clonal lineages colonizing children with and without AD, including 12 genotypes from A to L and 18 distinct sequence types (STs) from nine CCs (ST1/CC1, 188/1, 7618/1, 5/5, 1635/5, 8/8, 72/8, 1813/8, 15/15, 333/15, 2104/25, 30/30, 7743/30, 508/45, 7744/45, 97/97, 398/398 e 7742/398) (Figs 3 and 4). Four novel STs, single locus variant (SLV) from other STs, were identified: ST7618/CC1 that differs from ST188/CC1 by an alteration in the *aroE* allele; 7742 that differs from ST398 by an alteration in the *aroE* allele; 7743/CC30 that differs from ST30/CC30 by an alteration in the *aroE* allele; and 7744/CC45 that differs from ST45/CC45 by an alteration in the *pta* allele. Most of the *S. aureus* isolates belonged to CCs 30 (18.6%), 1 (15.7%), 5 (14.3%), and 398 (12.8%). Among MRSA isolates, the majority was related to STs/CCs 30/30 (50%) and 5/5 (31.2%).

Isolates of CCs 30 and 5 were not found in the children without AD, while 13 (44.8%) AD children were colonized by these lineages in the nares ($P = 0.034$) and 14 (46.7%) in the lesional skin ($P = 0.062$). In general, the same genotype was detected in the skin and nares (Figs 3 and 4). Most of the PVL-positive isolates belonged to the genotype C/CC30 (40%). It is also noteworthy that one of the four children with severe AD (child 2), who only had *S. aureus* detected in lesional skin, presented PVL-positive MRSA strains at all isolation sites, with those in the skin lesion and nostrils of ST30. For two isolates without PFGE analysis (151ad from the lesional skin of AD child 5 and 645ad from nares of AD child 22), the STs described were ST7742 (SLV of ST398) and 7744 (SLV of ST45/45), respectively.

## Discussion

Atopic dermatitis has been associated with cutaneous dysbiosis, and *S. aureus* is the predominant pathogen in this niche, especially in lesional skin [2]. Despite the importance of *S. aureus* colonization in AD, few studies evaluate its clonal lineages and colonization by CoNS, including its susceptibility to methicillin. Here, *Staphylococcus* spp. isolated from the nares and skin of 30 AD and 12 non-AD Brazilian children were characterized. We found a high colonization rate by polyclonal MRSA and methicillin-sensitive *S. aureus* (MSSA) isolates, as well as a high prevalence of methicillin-resistant staphylococci isolates among AD children.

In this study, colonization by only *S. aureus* on lesional skin was higher than on non-lesional skin of children with AD, data that agree with our findings on the mannitol fermenting colony count (suggestive of *S. aureus*), which was also high on lesional skin. Similarly, other groups have reported a high *S. aureus* count in skin with lesion when compared to skin without lesion and skin from individuals without the disease [8, 26]. In the present study, three of the four children with severe AD were colonized with *S. aureus* alone on lesional skin. It is known that *S. aureus* is associated with inflammatory events that can exacerbate AD symptoms [2, 3, 27–29]. Thus, the abundance of this microorganism in the skin, especially when it is alone, may be related to the disease severity [2, 26]. In fact, Blanchet-Réthoré et al. (2017) found a reduction of *S. aureus* in the skin of AD patients correlated with a reduction in SCORAD when evaluating the effect of applying a lotion containing a probiotic [30].

In turn, the high prevalence of *S. aureus* in children without AD in the present study (50%) could be explained by close contact with patients presenting the disease. The similarity between *S. aureus* isolates from AD patients and their relatives [31–33], as well as the similarity

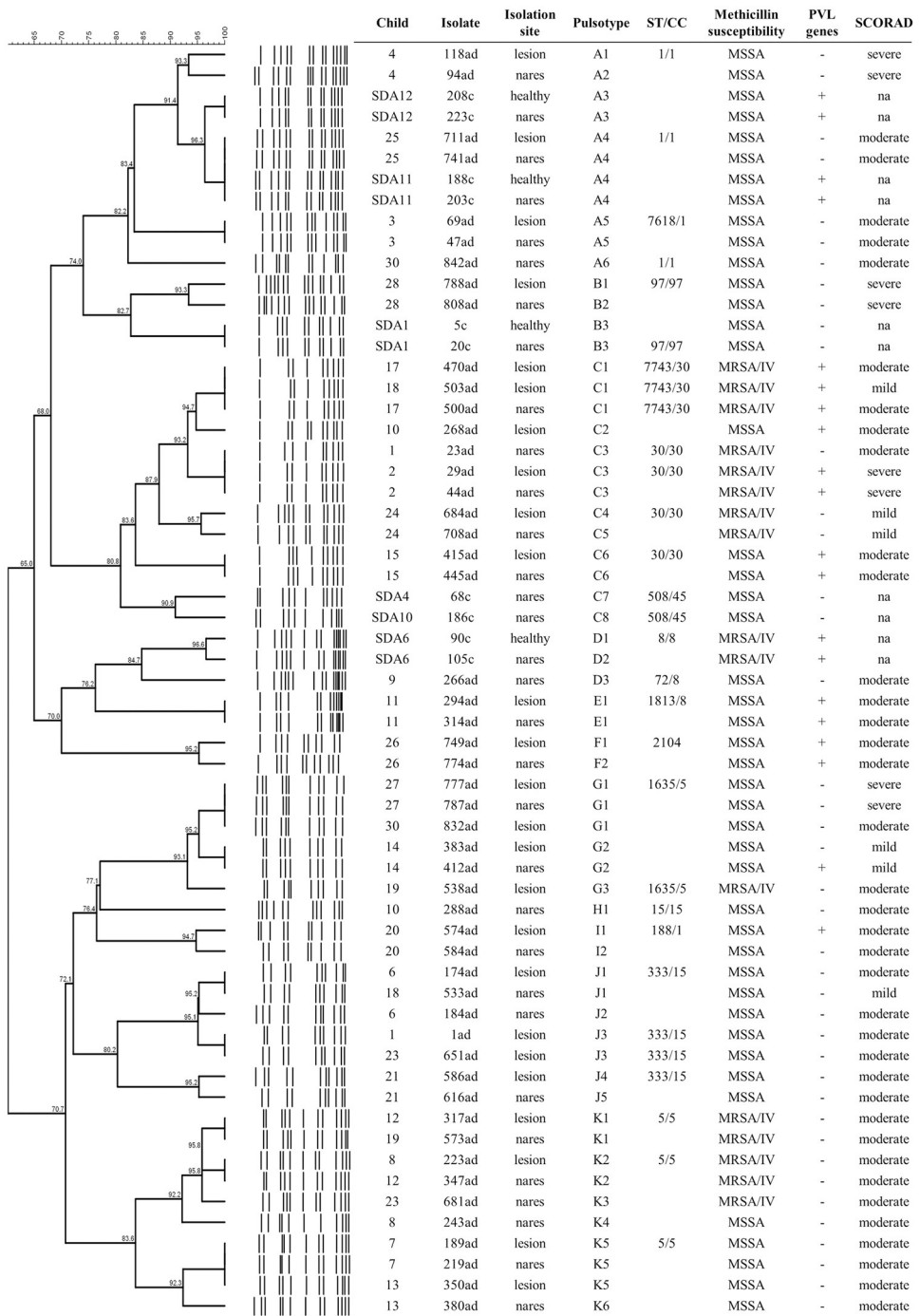

| Child | Isolate | Isolation site | Pulsotype | ST/CC | Methicillin susceptibility | PVL genes | SCORAD |
|---|---|---|---|---|---|---|---|
| 4 | 118ad | lesion | A1 | 1/1 | MSSA | - | severe |
| 4 | 94ad | nares | A2 | | MSSA | - | severe |
| SDA12 | 208c | healthy | A3 | | MSSA | + | na |
| SDA12 | 223c | nares | A3 | | MSSA | + | na |
| 25 | 711ad | lesion | A4 | 1/1 | MSSA | - | moderate |
| 25 | 741ad | nares | A4 | | MSSA | - | moderate |
| SDA11 | 188c | healthy | A4 | | MSSA | + | na |
| SDA11 | 203c | nares | A4 | | MSSA | + | na |
| 3 | 69ad | lesion | A5 | 7618/1 | MSSA | - | moderate |
| 3 | 47ad | nares | A5 | | MSSA | - | moderate |
| 30 | 842ad | nares | A6 | 1/1 | MSSA | - | moderate |
| 28 | 788ad | lesion | B1 | 97/97 | MSSA | - | severe |
| 28 | 808ad | nares | B2 | | MSSA | - | severe |
| SDA1 | 5c | healthy | B3 | | MSSA | - | na |
| SDA1 | 20c | nares | B3 | 97/97 | MSSA | - | na |
| 17 | 470ad | lesion | C1 | 7743/30 | MRSA/IV | + | moderate |
| 18 | 503ad | lesion | C1 | 7743/30 | MRSA/IV | + | mild |
| 17 | 500ad | nares | C1 | 7743/30 | MRSA/IV | + | moderate |
| 10 | 268ad | lesion | C2 | | MSSA | + | moderate |
| 1 | 23ad | nares | C3 | 30/30 | MRSA/IV | - | moderate |
| 2 | 29ad | lesion | C3 | 30/30 | MRSA/IV | + | severe |
| 2 | 44ad | nares | C3 | | MRSA/IV | + | severe |
| 24 | 684ad | lesion | C4 | 30/30 | MRSA/IV | - | mild |
| 24 | 708ad | nares | C5 | | MRSA/IV | - | mild |
| 15 | 415ad | lesion | C6 | 30/30 | MSSA | + | moderate |
| 15 | 445ad | nares | C6 | | MSSA | + | moderate |
| SDA4 | 68c | nares | C7 | 508/45 | MSSA | - | na |
| SDA10 | 186c | nares | C8 | 508/45 | MSSA | - | na |
| SDA6 | 90c | healthy | D1 | 8/8 | MRSA/IV | + | na |
| SDA6 | 105c | nares | D2 | | MRSA/IV | + | na |
| 9 | 266ad | nares | D3 | 72/8 | MSSA | - | moderate |
| 11 | 294ad | lesion | E1 | 1813/8 | MSSA | + | moderate |
| 11 | 314ad | nares | E1 | | MSSA | + | moderate |
| 26 | 749ad | lesion | F1 | 2104 | MSSA | + | moderate |
| 26 | 774ad | nares | F2 | | MSSA | + | moderate |
| 27 | 777ad | lesion | G1 | 1635/5 | MSSA | - | severe |
| 27 | 787ad | nares | G1 | | MSSA | - | severe |
| 30 | 832ad | lesion | G1 | | MSSA | - | moderate |
| 14 | 383ad | lesion | G2 | | MSSA | - | mild |
| 14 | 412ad | nares | G2 | | MSSA | + | mild |
| 19 | 538ad | lesion | G3 | 1635/5 | MRSA/IV | - | moderate |
| 10 | 288ad | nares | H1 | 15/15 | MSSA | - | moderate |
| 20 | 574ad | lesion | I1 | 188/1 | MSSA | + | moderate |
| 20 | 584ad | nares | I2 | | MSSA | - | moderate |
| 6 | 174ad | lesion | J1 | 333/15 | MSSA | - | moderate |
| 18 | 533ad | nares | J1 | | MSSA | - | mild |
| 6 | 184ad | nares | J2 | | MSSA | - | moderate |
| 1 | 1ad | lesion | J3 | 333/15 | MSSA | - | moderate |
| 23 | 651ad | lesion | J3 | 333/15 | MSSA | - | moderate |
| 21 | 586ad | lesion | J4 | 333/15 | MSSA | - | moderate |
| 21 | 616ad | nares | J5 | | MSSA | - | moderate |
| 12 | 317ad | lesion | K1 | 5/5 | MRSA/IV | - | moderate |
| 19 | 573ad | nares | K1 | | MRSA/IV | - | moderate |
| 8 | 223ad | lesion | K2 | 5/5 | MRSA/IV | - | moderate |
| 12 | 347ad | nares | K2 | | MRSA/IV | - | moderate |
| 23 | 681ad | nares | K3 | | MRSA/IV | - | moderate |
| 8 | 243ad | nares | K4 | | MSSA | - | moderate |
| 7 | 189ad | lesion | K5 | 5/5 | MSSA | - | moderate |
| 7 | 219ad | nares | K5 | | MSSA | - | moderate |
| 13 | 350ad | lesion | K5 | | MSSA | - | moderate |
| 13 | 380ad | nares | K6 | | MSSA | - | moderate |

**Fig 3. Dendrogram of the clonal profiles obtained after digestion with the restriction enzyme *Sma*I, and general characteristics of 61 *S. aureus* isolates from nasal and skin colonization of children with and without AD.** CC-Clonal complex; MRSA/IV- Methicillin-resistant *Staphylococcus aureus* presenting SCC*mec* IV; MSSA- Methicillin-sensitive *Staphylococcus aureus*; na- Not applicable; PVL- Panton-Valentine leukocidin; SDA- Child without atopic dermatitis; ST- Sequence-type.

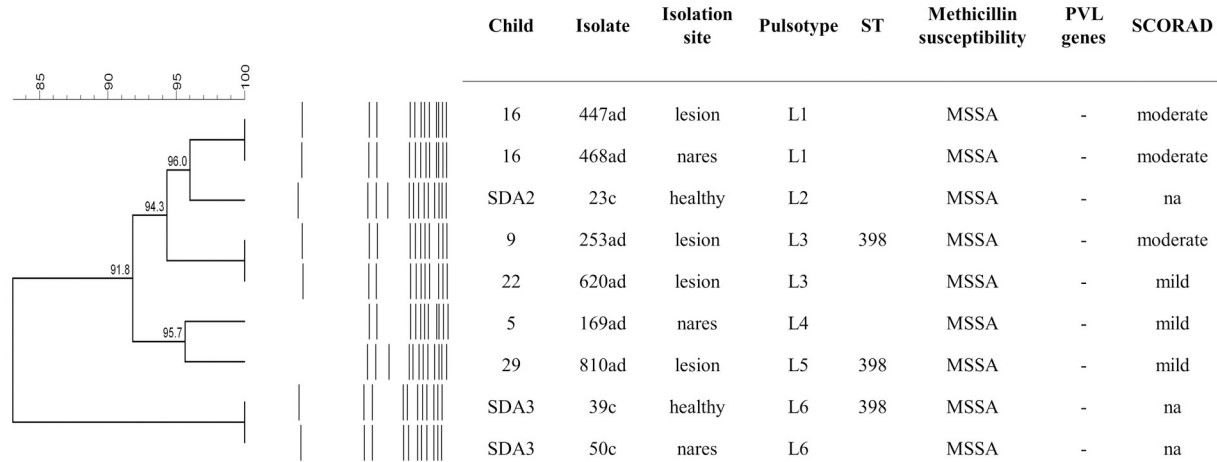

| Child | Isolate | Isolation site | Pulsotype | ST | Methicillin susceptibility | PVL genes | SCORAD |
|---|---|---|---|---|---|---|---|
| 16 | 447ad | lesion | L1 | | MSSA | - | moderate |
| 16 | 468ad | nares | L1 | | MSSA | - | moderate |
| SDA2 | 23c | healthy | L2 | | MSSA | - | na |
| 9 | 253ad | lesion | L3 | 398 | MSSA | - | moderate |
| 22 | 620ad | lesion | L3 | | MSSA | - | mild |
| 5 | 169ad | nares | L4 | | MSSA | - | mild |
| 29 | 810ad | lesion | L5 | 398 | MSSA | - | mild |
| SDA3 | 39c | healthy | L6 | 398 | MSSA | - | na |
| SDA3 | 50c | nares | L6 | | MSSA | - | na |

**Fig 4. Dendrogram of the clonal profiles obtained after digestion with the restriction enzymes *Apa*I and *Sma*I, and general characteristics of nine *S. aureus* isolates from nasal and skin colonization of children with and without AD.** MSSA- Methicillin-sensitive *Staphylococcus aureus*; na- Not applicable; PVL- Panton-Valentine leukocidin; SDA- Child without atopic dermatitis; ST- Sequence-type.

of *S. aureus* isolates found in different colonization sites of the same patient is also reported [31, 34–36]. Indeed, 20 (74.1%) children with AD were colonized by *S. aureus* of the same genotype in the lesional skin and nares, as well as all non-AD children who presented *S. aureus* at both sites evaluated. Thus, control measures for the eradication of *S. aureus* in AD patients, such as use of topical probiotics or bleach baths, should take into consideration different niches and close contacts, since they may serve as a focus for a new colonization episode or to keep the pathogen circulating in the environment. However, we emphasize that we do not advise the use of topical mupirocin in cases of colonization, even though a low prevalence of isolates with high levels of resistance was detected in this study, due to the possibility of emergence of antimicrobial resistance [37].

As previously demonstrated [4, 5, 32], *S. aureus* isolates colonizing the skin and nares of children with and without AD showed great clonal diversity. Here, we found at least 18 STs, and as in those studies [4, 5, 32], most isolates were of STs 30, 5, and 1, from the CC 30, 5 and 1, respectively. These are major lineages associated with community settings and are also detected in Brazilian studies colonizing and infecting hospitalized patients [38–42]. In addition, four STs that have never been described before were detected (STs 7618, 7742, 7743 and 7744). Similar to this study, Cavalcante and colleagues (2015) also found an MSSA isolate of a novel ST described as a SLV of ST188 in a child with AD [5]. It is also worth noting that nine isolates were not cut by the restriction enzyme *Sma*I on PFGE with representatives of the ST398 clustered among them, which are reported presenting this feature [43–45]. Although this lineage was initially described in livestock-associated MRSA isolates [43], it has been increasingly detected worldwide colonizing and infecting humans without animal exposure [46], including among AD patients [4, 47, 48]. The new STs detected in the present study, as well as the sporadic lineages, reinforce the diversity of *S. aureus* isolates colonizing AD children and the need to understand their role in disease severity. Moreover, the distribution of clonal lineages observed in AD individuals seems to resemble the epidemiology of the place where the studies are conducted.

A high diversity of *S. aureus* identified in our study could be also explained by the amount of MSSA isolates investigated, since SCC*mec* acquisition is reported to be restricted to certain clonal lineages [49]. Of note is the fact that the MRSA isolates characterized in the present

study were mainly from STs 30 and 5, which were only found among isolates from AD children. It was previously reported that most MRSA isolates from infected skin of Brazilian pediatric patients with AD were USA1100/ST30 and USA800/ST5, and both carried different virulence genes, including the PVL genes [4]. Even though some studies have not observed a clear relationship between *S. aureus* clonal lineages colonizing AD patients and disease severity [34, 50], it is known that certain lineages are associated with specific characteristics, such as presence of virulence genes, biofilm formation, acquisition of resistance mechanisms, among others [4, 49, 51, 52]. These findings highlight the importance of monitoring colonization by resistant isolates in this group, especially of virulent lineages, since colonization by MRSA may be associated with reduced microbial diversity of the skin [53] and presentation of more severe forms of the disease [37].

Studies evaluating the methicillin susceptibility profile in *S. aureus* isolated from AD patients show different rates (ranging between 0 and 25%) of MRSA isolates [48, 53–58], without a clear correlation as to whether patients with AD are more colonized by MRSA than individuals without the disease [54, 55]. The prevalence of MRSA colonization may vary according to epidemiological aspects of where studies are conducted [5, 53–59], and there are also reports of individuals who face worse socioeconomic conditions tend to be colonized by MRSA more frequently [37, 60]. While some authors did not find any AD patients in the southern region of Brazil colonized by MRSA [57, 59], Cavalcante and colleagues (2015) found that 23% of pediatric AD patients attending a dermatology unit in Rio de Janeiro were colonized by MRSA in the nares and/or infected skin, with most of these isolates carrying SCC*mec* IV [5]. Similarly, in this study, 30% of AD patients presented MRSA/SCC*mec* IV in at least one of the sites evaluated, and both AD patients and those without the disease presented colonization by MRSA isolates.

Although *S. aureus* is arguably a relevant pathogen in AD, the role of other staphylococcal species in AD is still poorly understood. In this study, at least one staphylococcal species was detected in all investigated sites, and, in most of them, more than one species was isolated at each investigated site. In fact, studies evaluating the skin microbiome of people with and without AD have detected the presence of different staphylococcal species in association [2, 8, 26]. In our study, the most isolated species from skin in the AD group were *S. aureus* and *S. epidermidis*, as previously shown by another study from Soares and colleagues (2013) [61]. Unlike our results, *S. epidermidis* and *S. haemolyticus* were the most frequently detected CoNS species colonizing the skin of adult AD patients in a Polish study [62]. In general, studies assessing *Staphylococcus* spp. colonization in people with and without AD point to a varied colonization by CoNS in these individuals [7, 26, 53, 63, 64]. Colonization by *S. capitis* was correlated with greater AD severity in studies conducted in Denmark and South Africa [26, 65]. Similar to the present work, a lower prevalence of *S. hominis* in people with AD and a negative association between its abundance and AD severity is reported [7, 26, 53, 64]. Therefore, it is possible that colonization by certain bacterial species might contribute to the worsening of the disease, while other species, such as *S. hominis*, play a protective role in AD.

Despite their reported beneficial role, it is worth noting that CoNS are notorious for their antimicrobial resistance and are considered reservoirs of transferable resistance genes, such as SCC*mec* [66]. Most of the isolates evaluated in our study were methicillin-resistant, mainly *S. epidermidis*, *S. haemolyticus* and *S. hominis*. Only one child from each group colonized by SCN did not present resistant isolates in at least one of the investigated sites. A high prevalence of MR-CoNS from nasal colonization had already been reported by our group in neonates in an intensive care unit, in which 90.3% of *S. haemolyticus* and 87.5% of *S. epidermidis* isolates showed this characteristic [67]. Byrd et al. (2017) also reported a predominance of methicillin-resistant *S. epidermidis* colonizing patients with AD [2]. However, only 1.2% of the CoNS

isolates colonizing the lesional skin from South Korean AD patients presented resistance to methicillin [50]. To the best of our knowledge, this is the first report of a high prevalence of MR-CoNS of different species colonizing the skin and nares of AD patients, highlighting the importance of resistance monitoring in this population that is frequently subjected to antimicrobial therapy due to recurrent bacterial infections.

Our results should be evaluated with caution since the sample size was small for both groups and a culture-dependent method was used, which may not have favored the identification of less abundant species. Furthermore, due to the inclusion criteria of the study which required a skin lesion in the antecubital fossa and the lack of use of systemic immunosuppressors, most of the patients presented moderate AD, which impaired the establishment of relationship between the results and disease severity.

## Conclusion

High colonization by polyclonal *S. aureus* isolates was found among children with AD, while *S. hominis* was more frequent among non-AD children, reinforcing that dysbiosis is an important aspect of this disease. The high number of methicillin-resistant isolates demonstrates the need for continued monitoring of this population who are at increased risk for recurrent infections, especially when considering empirical antibiotic therapy for the treatment of skin infections in patients with AD.

## Supporting information

**S1 Table. Description of the PCR conditions used to confirm staphylococcal identification.** (DOC)

**S2 Table. General characteristics of the 42 children included in the study.** Light grey-Absent; Dark grey- Present; SDA- Child without atopic dermatitis; SCORAD- *Scoring atopic dermatitis*; SCORAD 1- Mild; SCORAD 2- Moderate; SCORAD 3- Severe; na- Not applicable; F- Female; M- Male; Sa- *S. aureus;* Sc- *S. capitis;* Se- *S. epidermidis;* Sh- *S. haemolyticus;* Sn- *S. hominis;* Ss- *S. saprophyticus;* O(s)- Another coagulase-negative *Staphylococcus* spp.; MRSA-Methicillin-resistant *S. aureus;* Sa PVL+- *S. aureus* presenting the Panton-Valentine leukocidin genes; MR-CoNS- Methicillin-resistant coagulase-negative *Staphylococcus;* Child 27*- Child with atopic dermatitis without a swab from non-lesional skin. (DOC)

## Acknowledgments

We thank the children and the research team that participated in this study.

## Author Contributions

**Conceptualization:** Lorrayne Cardoso Guimarães, Dennis de Carvalho Ferreira, Kátia Regina Netto dos Santos.

**Data curation:** Lorrayne Cardoso Guimarães.

**Formal analysis:** Lorrayne Cardoso Guimarães.

**Funding acquisition:** Dennis de Carvalho Ferreira, Kátia Regina Netto dos Santos.

**Investigation:** Lorrayne Cardoso Guimarães, Maria Isabella de Menezes Macedo Assunção, Tamara Lopes Rocha de Oliveira.

**Methodology:** Lorrayne Cardoso Guimarães, Dennis de Carvalho Ferreira, Kátia Regina Netto dos Santos.

**Project administration:** Dennis de Carvalho Ferreira, Kátia Regina Netto dos Santos.

**Resources:** Simone Saintive, Eliane de Dios Abad, Ekaterini Simoes Goudouris, Evandro Alves do Prado, Dennis de Carvalho Ferreira, Kátia Regina Netto dos Santos.

**Supervision:** Dennis de Carvalho Ferreira, Kátia Regina Netto dos Santos.

**Validation:** Lorrayne Cardoso Guimarães, Tamara Lopes Rocha de Oliveira, Fernanda Sampaio Cavalcante, Dennis de Carvalho Ferreira, Kátia Regina Netto dos Santos.

**Visualization:** Lorrayne Cardoso Guimarães, Dennis de Carvalho Ferreira, Kátia Regina Netto dos Santos.

**Writing – original draft:** Lorrayne Cardoso Guimarães.

**Writing – review & editing:** Lorrayne Cardoso Guimarães, Fernanda Sampaio Cavalcante, Simone Saintive, Eliane de Dios Abad, Ekaterini Simoes Goudouris, Evandro Alves do Prado, Dennis de Carvalho Ferreira, Kátia Regina Netto dos Santos.

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
