## [Decision Letter · Decision Letter 0]

26 Sep 2022

PONE-D-22-24445Polyclonal methicillin-resistant and -sensitive Staphylococcus aureus isolates in skin and nares from Brazilian children with atopic dermatitisPLOS ONE

Dear Dr. dos Santos,

Thank you for submitting your manuscript to PLOS ONE. After careful consideration, we feel that it has merit but does not fully meet PLOS ONE’s publication criteria as it currently stands. Therefore, we invite you to submit a revised version of the manuscript that addresses the points raised during the review process.

We look forward to receiving your revised manuscript.

Kind regards,

Feroze Kaliyadan, M.D.

Academic Editor

PLOS ONE

Journal Requirements:

Reviewers' comments:

Reviewer's Responses to Questions

**Comments to the Author**

1. Is the manuscript technically sound, and do the data support the conclusions?

Reviewer #1: Yes

Reviewer #2: Yes

2. Has the statistical analysis been performed appropriately and rigorously? 

Reviewer #1: Yes

Reviewer #2: Yes

3. Have the authors made all data underlying the findings in their manuscript fully available?

Reviewer #1: Yes

Reviewer #2: Yes

4. Is the manuscript presented in an intelligible fashion and written in standard English?

Reviewer #1: Yes

Reviewer #2: Yes

5. Review Comments to the Author

Reviewer #1: 1. Can the title be rephrased as: Methicillin-resistant and methicillin sensitive Staphylococcus aureus isolates from skin and nares of Brazilian children with atopic dermatitis demonstrate high level of clonal diversity. This would make it more appealing and interesting.

2. Highlight in the introduction part: What is the significance of the study? What is the significance and relevance of molecular characterization of MRSA in AD?

3. The paper needs grammar check. A few grammatical errors are found in the paper.

e.g.: page 3, line 60-61

"The presence of strains carrying genes for toxins is reported, which may be associated with triggering, maintaining, and increasing of the inflammatory process".

This sentence could be framed as below.

"The presence of virulence and toxin genes has been reported among staphylococcal isolates from AD, which may be associated with triggering, sustaining, and amplifying the inflammatory process".

The authors are advised to have the paper checked by a native English speaker.

4. Any mention of abbreviation should be with the full term for the first time and the abbreviations written in the parenthesis. Subsequently only the abbreviation can be used thereof. This has not been followed.

e.g.:

a. page 3, line 65. MRSA is mentioned in parenthesis for community methicillin-resistant

b. page 4, line 88. IPPMG

5. There are some instances of incomplete sentences.

e.g.: page 5, line 104-105.

"Unable to collect a swab from the non-lesional skin of one AD child".

This could be written as, " We were unable to collect a swab from the non-lesional skin of one AD child.

Reviewer #2: Well conducted study. The findings, especially regarding the high prevalence of MR-CoNS of different species colonizing the skin and nares of AD patients are valuable.

Have the following clarifications/ comments:

-Were there any frank infections like cellulitis/ abscesses/ impetigo noted in the study patients.

-Severity scoring (clinical severity) would perhaps have added more value

-In the children without AD with Staph aureus colonisation, were there any risk factors present for acquiring the same.

6. PLOS authors have the option to publish the peer review history of their article (what does this mean?). If published, this will include your full peer review and any attached files.

Reviewer #1: **Yes: **Dr. Sayed. A. Quadri M.D, Division of Microbiology and Immunology, Department of Biomedical Sciences, College of Medicine, King Faisal University, Al-Ahsa, Saudi Arabia

Reviewer #2: No

---

## [Author Response · Author response to Decision Letter 0]

14 Oct 2022

Journal Requirements:

>>Done. 

“Author contributions” section removed. 

Tables 1 and 2 formatted according to the style requirements: text center-aligned; font size changed; extra spaces removed.

Position of the legends of all figures formatted.

Position of the Table 1 footnote formatted.

>>We removed the phrase “data not shown” (line 216).

>>We changed reference 27 for another study as the previously one presented an erratum.

The reference 21 also presents an erratum. However, we did not change this reference since it is largely used for SCCmec typing and it has 189 citations according to Pubmed.

Review Comments to the Author

Reviewer #1: 

1. Can the title be rephrased as: Methicillin-resistant and methicillin sensitive Staphylococcus aureus isolates from skin and nares of Brazilian children with atopic dermatitis demonstrate high level of clonal diversity. This would make it more appealing and interesting.

>>Done.

2. Highlight in the introduction part: What is the significance of the study? What is the significance and relevance of molecular characterization of MRSA in AD?

>>We added two sentences in the last paragraph of the introduction answering those questions (lines 82 and 83; 87-90).

3. The paper needs grammar check. A few grammatical errors are found in the paper.

e.g.: page 3, line 60-61

"The presence of strains carrying genes for toxins is reported, which may be associated with triggering, maintaining, and increasing of the inflammatory process".

This sentence could be framed as below.

"The presence of virulence and toxin genes has been reported among staphylococcal isolates from AD, which may be associated with triggering, sustaining, and amplifying the inflammatory process".

The authors are advised to have the paper checked by a native English speaker.

>>Done.

4. Any mention of abbreviation should be with the full term for the first time and the abbreviations written in the parenthesis. Subsequently only the abbreviation can be used thereof. This has not been followed.

e.g.:

a. page 3, line 65. MRSA is mentioned in parenthesis for community methicillin-resistant

b. page 4, line 88. IPPMG

>>Done.

5. There are some instances of incomplete sentences.

e.g.: page 5, line 104-105.

"Unable to collect a swab from the non-lesional skin of one AD child".

This could be written as, " We were unable to collect a swab from the non-lesional skin of one AD child.

>>Done.

Reviewer #2: 

Well conducted study. The findings, especially regarding the high prevalence of MR-CoNS of different species colonizing the skin and nares of AD patients are valuable.

Have the following clarifications/ comments:

-Were there any frank infections like cellulitis/ abscesses/ impetigo noted in the study patients.

>>As stated in the lines 102 and 103, patients with any sign of infection were not included in the study.

-Severity scoring (clinical severity) would perhaps have added more value

>>The severity of the disease was assessed in the day of the sample collection using the SCORAD index (lines 100 and 101). We presented the distribution of the patients according to the SCORAD in lines 167 and 168. We also mention that some patients with moderate and severe AD only had S. aureus recovered from skin sites (lines 211-214; 266-268; 301-305).

-In the children without AD with Staph aureus colonisation, were there any risk factors present for acquiring the same.

>>The children without AD were siblings of AD children (lines 98-100), which could be a risk factor for acquiring S. aureus, since these children have close contact with AD patients (lines 306-315).

---

## [Editor Report · Decision Letter 1]

18 Oct 2022

Methicillin-resistant and methicillin-sensitive Staphylococcus aureus isolates from skin and nares of Brazilian children with atopic dermatitis demonstrate high level of clonal diversity

PONE-D-22-24445R1

Dear Dr. dos Santos,

We’re pleased to inform you that your manuscript has been judged scientifically suitable for publication and will be formally accepted for publication once it meets all outstanding technical requirements.

Kind regards,

Feroze Kaliyadan, M.D.

Academic Editor

PLOS ONE

:

---

## [Editor Report · Acceptance letter]

24 Oct 2022

PONE-D-22-24445R1 

Methicillin-resistant and methicillin-sensitive *Staphylococcus aureus* isolates from skin and nares of Brazilian children with atopic dermatitis demonstrate high level of clonal diversity 

Dear Dr. dos Santos:

I'm pleased to inform you that your manuscript has been deemed suitable for publication in PLOS ONE. Congratulations! Your manuscript is now with our production department. 

Kind regards, 

on behalf of

Dr. Feroze Kaliyadan 

Academic Editor

PLOS ONE